# Availability, prices and affordability of essential medicines for treatment of diabetes and hypertension in private pharmacies in Zambia

Andrea Hannah Kaiser[1], Lindsey Hehman[2], Birger Carl Forsberg[3]*, Warren Mukelabai Simangolwa[2], Jesper Sundewall[4]

**1** Karolinska Institutet, Stockholm, Sweden, **2** Clinton Health Access Initiative, Inc. (CHAI), Lusaka, Zambia, **3** Department of Public Health Sciences, Karolinska Institutet, Stockholm, Sweden, **4** HEARD, University of Kwazulu-Natal and Division of Social Medicine and Global Health, Lund University, Lund, Sweden

* birger.forsberg@ki.se

**Data Availability Statement:** All relevant data are within the manuscript and its Supporting Information files.

## Abstract

### Objectives

To explore availability, prices and affordability of essential medicines for diabetes and hypertension treatment in private pharmacies in three provinces of Zambia.

### Methods

A cross-sectional survey was conducted in 99 pharmacies across three Zambian provinces. Methods were based on a standardized methodology by the World Health Organization and Health Action International. Availability was analysed as mean availability per pharmacy and individual medicine. Median prices were compared to international reference prices and differences in price between medicine forms (original brand or generic product) were computed. Affordability was assessed as number of days' salaries required to purchase a standard treatment course using the absolute poverty line and mean per capita provincial household income as standard. An analysis identifying medicines considered both available and affordable was conducted.

### Results

Two antidiabetics and nine antihypertensives had high-level availability (≥80%) in all provinces; availability levels for the remaining surveyed antidiabetics and antihypertensives were largely found below 50%. Availability further varied markedly across medicines and medicine forms. Prices for most medicines were higher than international reference prices and great price variations were found between pharmacies, medicines and medicine forms. Compared to original brand products, purchase of generics was associated with price savings for patients between 21.54% and 96.47%. No medicine was affordable against the absolute poverty line and only between four and eleven using mean per capita provincial

**Funding:** AHK received funding from the Clinton Health Access Initiative (CHAI, https://clintonhealthaccess.org). The funders had no role in study design, data collection and analysis, decision to publish, or preparation of the manuscript.

**Competing interests:** The authors have declared that no competing interests exist.

incomes. Seven generics in Copperbelt/Lusaka and two in Central province were highly available and affordable.

## Conclusions

The study showed that the majority of surveyed antidiabetic and antihypertensive medicines was inadequately available (<80%). In addition, most prices were higher than their international reference prices and that treatment with these medicines was largely unaffordable against the set affordability thresholds. Underlying reasons for the findings should be explored as a basis for targeted policy initiatives.

## Introduction

In line with its pledge to achieve Universal Health Coverage by 2030, the government of the Republic of Zambia is planning to establish a National Health Insurance (NHI) scheme [1]. The corresponding law was passed in 2018 and mandates the National Health Insurance Management Authority to develop a comprehensive health benefits package (HBP), taking special account of the national burden of disease and national public health priorities [1,2].

One such priority area is the growing burden of noncommunicable diseases (NCDs), primarily diabetes mellitus (diabetes) and hypertension [3,4]. Diabetes is a metabolic condition characterized by excessive blood glucose levels [3]. Hypertension is an arterial disease with persistently elevated blood pressure [4]. Both are chronic, progressive diseases and can lead to severe complications including cardiovascular disease and kidney failure and premature deaths if not appropriately managed and controlled [3,4]. Primary prevention, early diagnosis and adequate treatment are key to avoid such dire consequences and associated economic losses [3,4]. However, diabetes and hypertension often go undetected in lower and middle-income countries (LMICs)–including Zambia [3–5], due to prevailing financial and human resource constraints, insidious onset of the diseases and lack of awareness by those affected. With prevalence estimates of 7.6% for diabetes and 41.6% for hypertension, both accounted for almost half of Zambia's NCDs burden already in 2013 and were additionally among the 10 leading causes of admission to and mortality in public health facilities from 2009 to 2015 [1,5,6]. Data from the central health management information system (HMIS) indicate that the prevalence has been rising considerably since, a trend which is expected to continue in coming decades [7].

Long-term, often lifelong treatment requiring continuous medication is common for diabetic and hypertensive patients, which makes access to essential medicines (EMs) vital for treatment success and, ultimately, survival [3,4]. Effective antidiabetic and antihypertensive medicines with good margins of safety have been developed and been available for many years, such as oral hypoglycaemics and insulin for diabetes [3] and diuretics, beta blockers and angiotensin converting enzyme inhibitors for hypertension [4].

However, EMs for diabetes and hypertension have been found to be intermittent in supply, unavailable and unaffordable in studies across LMICs [8–10]. Zambia has a policy of providing medicines free of charge for public primary health care [11], yet due to resource limitations, medicines are available irregularly. For instance, a 2017 study in 15 public facilities in Lusaka found availability levels of only 44.7% to 58.2% and stock out rates between 69% and 92% for a number of antidiabetics and antihypertensives [12]. When public facilities are unable to dispense required medicines, patients have to purchase them out of their "own pocket" at

prevailing prices in the private sector, where availability levels are frequently higher [9,13,14]. Figures for private sector patient prices in Zambia vary widely and no reliable estimates are available, yet they have generally been observed to be high [14,15]. In this light, the 2014 Zambian Household Expenditure and Utilization Survey found that 42% of out-of-pocket payments (OOPs) incurring in rural and urban populations of all income groups were related to medicine expenses [16]. Although these estimates do not relate to diabetes or hypertension in particular, they are nonetheless indicative of the economic burden these chronic conditions can impose on individuals and households [3,4].

With the planned introduction of the NHI, cooperation and complementarity of health service delivery, including medicine provision, between the public and private sector are foreseen [17]. A solid understanding of availability, prices and affordability of antidiabetic and antihypertensive medicines in both sectors is hence essential to enable planning and realizing an effective intersectoral cooperation. There is, however, a paucity of reliable data and very little research has been carried out into the specific scope of the challenges regarding these medicines. Against this background, the aim of this study was to systematically assess availability, prices and affordability of EMs for diabetes and hypertension treatment in private pharmacies in three provinces of Zambia.

## Methods

A descriptive cross-sectional survey of availability, prices and affordability of essential antidiabetic and antihypertensive medicines was conducted. Study design and methods were adapted from a standardized methodology for systematic facility-based surveys developed by the World Health Organization (WHO) and Health Action International (HAI) [18].

### Survey scope

The three adjacent provinces Central, Copperbelt and Lusaka were purposively selected as survey scope. They account for about 22% of the total surface area and 43% of the total population (2015) in Zambia [19] and are home to 89% (191 of 214) of all private retail pharmacy businesses (hereinafter referred to as "pharmacies") that have official operating licenses from the Zambian Medicines Regulatory Authority (ZAMRA) [20]. Copperbelt and Lusaka are Zambia's most urban and densely populated provinces, while Central is more sparsely inhabited and considered predominantly rural [21]. The rationale behind choosing this study area was based on three main components. Firstly, Zambia is geographically vast and has one of the lowest population densities in Sub-Saharan Africa [21], in which cases the applied WHO/HAI methodology recommends conducting a provincial survey or a series of provincial-based surveys [18]. Secondly, 2018 routine data collected from the central HMIS showed that Central, Copperbelt and Lusaka province had the highest utilization of public hospitals by diabetes and hypertension patients seeking treatment of all provinces. Thirdly, they were found to have the highest shares of total household OOPs of all provinces in the 2014 Zambian Household Expenditure and Utilization Survey, a large proportion of which relates to medicine spending [16].

### Selection of pharmacies

A sample of 99 pharmacies was surveyed based on the 2018 register of licensed private retail pharmacies published by ZAMRA [20]. Due to the small total number of registered pharmacies in Central province (n = 6), these were surveyed exhaustively. In Lusaka and Copperbelt province, pharmacies were selected by simple random sampling based on methods developed by Measure Evaluation [22]. The applied formula determined the sample size based on the

minimum acceptable margin of error of the estimate, the level of confidence desired and the approximate prevalence of the parameter of interest in the facility population [22]. As the total population of pharmacies was predetermined and relatively small in both provinces (N = 149 in Lusaka and N = 36 in Copperbelt province), the resulting n was greater than 5% of the total, and the sample sizes were correspondingly adjusted with a Finite Population Correction factor [22]. Resulting sample sizes were n = 65 for Lusaka and n = 28 for Copperbelt province. Whilst retaining the numbering of the ZAMRA register, pharmacies located in Copperbelt and Lusaka province were extracted and entered into Microsoft Excel spreadsheets. A random number sequence was then created to randomly select pharmacies corresponding to the sample sizes. In addition, 10% of the sample sizes were selected as backup pharmacies; no backups could be selected in Central province.

## Selection of medicines

A standardized data collection form was developed to ensure consistency of data collection across pharmacies and provide for maximum data accuracy and reliability. A total of 32 medicines including 7 antidiabetics and 25 antihypertensives were included in the survey. Medicine selection was based on the 2017 WHO Model List of EMs [23] and the 2017 Zambia Essential Medicines List (ZEML) [24] and further cross-checked against the Zambia National Formulary and global diabetes and hypertension guidelines to ensure completeness [25–27]. Selection was finalised in consultation with a practicing Zambian pharmacist and officials from Medical Stores Limited [28].To be included in the final survey list, applied inclusion criteria were that medicines needed to have international reference prices (IRPs) in the Drug Price Indicator Guide by Management Sciences for Health (MSH) [29] and further to be authorised for sale in Zambia based on the 2018 ZAMRA register of market authorisations of medicines for human use [30]. The MSH IRP set is based upon recent procurement prices for generics offered to LMICs by international suppliers and provides a benchmark of reasonable medicine prices [18,29]. Dosage form and strength were specified as per the essential medicine lists (EMLs). Both single-ingredient and fixed-dose combination products were considered, provided that all the requirements for inclusion in the survey were met. Furthermore, as no standard for package sizes has been adopted in Zambia, these were assumed to correspond to a standard treatment course (STC) for chronic diseases, defined as 30 days [18]. To verify effectiveness of the data collection approach and accuracy of the tool, local pilot surveys were conducted at two pharmacies in Lusaka (data not included in final analysis).

## Data collection

All data were collected by a single trained investigator between February 20 and March 15, 2019. In keeping with the descriptive cross-sectional study design, data on all variables were collected at a single time point for each pharmacy. A print version of the data collection form was prepared with unique identifiers for all pharmacies and backups. Information on medicine availability and prices was recorded with the aid of skilled pharmacy staff present during the survey visits, i.e. pharmacists or pharmacy technicians. Two forms were surveyed for each medicine, namely the original brand product (OBP), more specifically the brand-name proprietary product that first obtained worldwide market authorisation, and the lowest-priced generic (LPG), defined as the cheapest generic equivalent found at each pharmacy at the time of the survey [18]. Generic insulin refers strictly speaking to biosimilar insulin [31] but was included under LPGs for reasons of simplicity. Data were only collected for specified strength (s) and dosage form(s). To be classified "available", medicines had to be physically seen by the researcher. Price data were only recorded for medicines in stock and taken from product

labels, price schedules, computer databases or, in the absence thereof, based upon verbal communication with staff. Legibility and completeness of data collection forms were verified before leaving pharmacies. Obtained data were transferred into a Microsoft Excel workbook; a double-entry technique was used to ensure accuracy of data entry. In informal discussions during the data collection, interesting and relevant information was shared by participating pharmacy staff, which was documented as field notes by the researcher. Information providing additional insights into the topic of study that was deemed to merit inclusion in the article was used to inform the discussion.

### Data analysis

Analyses were carried out separately for the three provinces due to the differences in both the sample sizes and provinces' mean per capita monthly household incomes, which were used as income measures in the affordability analysis.

### Assessment of availability

Availability was assessed as the percentage of pharmacies where the medicine was found at the time of the survey and as mean percentage availability across the basket of selected antidiabetics and antihypertensives. In addition, mean percentage availability of all survey medicines was calculated for each pharmacy. Following the approach of previous studies [32–34], 5 categories were established to classify availability: absent (0%), very low (<30%), low (30-<50%), fairly high (50-<80%), high (≥80%).

### Assessment of prices

Median prices were calculated in local currency (Zambia Kwacha) and converted to US$ using the Bank of Zambia's Zambia Kwacha-US$ exchange rate [35]. The mean of the daily average exchange rates during the data collection period was used. On this basis, median price ratios (MPRs), i.e. a ratio of the local price to MSH IRPs, were calculated for medicines available at a minimum of 4 pharmacies within each province [18,29]. Interquartile ranges (IQRs) were computed to estimate the extent of price variation across pharmacies and medicines [18]. Following a 2005 WHO study on chronic disease medication, MPRs ≤ 2.5 were considered acceptable; values above this threshold were judged as excessive local prices [32]. Moreover, for medicines for which both OBPs and LPGs were found in at least 4 pharmacies per province, the potential saving for a customer through opting for the generic product was calculated as a saving ratio: *Saving ratio = (price$_{OBP}$−price$_{LPG}$) / (price$_{OBP}$ * 100%)*.

### Assessment of affordability

Treatment affordability was defined as the ability to pay for a standard treatment course for diabetes or hypertension, expressed as the number of days' income needed for its purchase [18]. A medicine was classified affordable if not exceeding a maximum of one day' salary for a STC [18]. The daily salary was estimated from Zambia's absolute poverty line or from the mean per capita monthly household income in the provinces [19]. Values were taken from the 2015 Living Conditions Monitoring Survey Report [19], inflated to 2018-Zambia Kwacha by use of Zambian consumer price indices (latest available data) [36,37] and converted to 2018-US$ using the same exchange rate as for the price assessments. The absolute poverty line lay at US$13.15 per month and average monthly incomes in the provinces were US$24.10 (Central), US$46.25 (Copperbelt) and US$48.85 (Lusaka). Defined daily dosages were used to approximate STCs and were taken from the WHO Collaborating Centre for Medicine

Statistics Methodology [38]. For medicines available in at least 4 pharmacies per province, STCs were multiplied with median medicine prices to obtain costs of a 30-day treatment. In line with target nine in the WHO Global monitoring framework for NCDs, stipulating an 80% availability of affordable EMs in public and private facilities [39], a comprehensive analysis of both medicine availability and affordability was additionally conducted. In this way, antidiabetics and antihypertensives considered available (available in ≥80% of pharmacies surveyed) and affordable (STC ≤ one day's salary) were identified.

### Ethics statement

Before commencing data collection, the study was officially reviewed by Clinton Health Access Initiative's internal Scientific and Ethical Review Committee (SERC). The committee deemed the study as non-human subject research as no data were collected about an individual but only on medicine availability and prices (despite the interaction with human subjects to get information during the data collection at the pharmacies following the obtainment of verbal informed consent). Consequently, no formal ethical approval was sought from the University of Zambia's Biomedical Research Ethics Committee. Verbal informed consent was obtained from all participants included in the study. Pharmacy managers and/or staff were provided all relevant information including study background, purpose, methods, potential risks and benefits as well the voluntariness of participation before consenting to data collection. Information was provided as a written leaflet in Zambia's official language English. The leaflet had been developed together with a Zambian pharmacist.

## Results

Of the 32 selected medicines, 12 had no OBP registered with ZAMRA and were hence excluded from the survey. The survey medicines thus comprised 14 OBPs (Diabetes: 6; Hypertension: 8) and 32 LPGs (Diabetes: 7; Hypertension: 25). Data on prices and availability of these were collected from 99 pharmacies in the selected provinces Central, Copperbelt and Lusaka, including eight backups because pharmacies of the primary sample were not willing to participate (n = 2), not found in the location indicated (n = 4) or no longer in operation (n = 2). With regards to the distribution of the pharmacies in the study area, the surveyed pharmacies were invariably located in urban areas. Even in Central province, all surveyed pharmacies were located in the province's major town with over 200,000 inhabitants.

### Availability

Mean availability across all surveyed antidiabetic and antihypertensive medicines in the surveyed pharmacies was low (30-<50%) in all provinces (Table 1). Assessing availability for only medicines included in the WHO Model EML [23] showed slightly higher mean availability, yet estimates still fell into the low-level availability category. In total, 6 pharmacies had high-level stock availability (≥80%) of all surveyed medicines.

Table 2 shows the availability profile of the surveyed medicines. Overall mean availability across the baskets of antidiabetics and antihypertensives was low (30-<50%) in all provinces. Regarding availability of individual medicines, high-level availability (≥80%) in all provinces was found for 2 generic antidiabetics and 9 antihypertensives. The majority of the remaining medicines was available at low (30-<50%) and very low (<30%) levels. Individual medicine availability was further found to vary with medicine forms. LPGs were more readily available than OBPs in all provinces; no single antihypertensive OBP was found in Central province. Regarding insulin in particular, all types had low or very low availability for both OBPs and biosimilars. The least available antihypertensives were injectable Hydralazine and Metoprolol

**Table 1. Availability of all medicines in the surveyed pharmacies.**

| Province | | Central (n = 6) | | Copperbelt (n = 28) | | Lusaka (n = 65) | |
|---|---|---|---|---|---|---|---|
| Mean availability all surveyed medicines | | 35.14% | | 42.47% | | 42.01% | |
| Mean availability WHO EML medicines | | 41.11% | | 46.90% | | 44.62% | |
| Availability category | | No. | % | No. | % | No. | % |
| <30% | Very low | 3 | 50.00 | 5 | 17.86 | 19 | 29.23 |
| 30-<50% | Low | 3 | 50.00 | 16 | 57.14 | 25 | 38.46 |
| 50-<80% | Fairly high | 0 | 0.00 | 5 | 17.86 | 17 | 26.15 |
| ≥80% | High | 0 | 0.00 | 2 | 7.14 | 4 | 6.15 |

EML = essential medicines list; n = number of surveyed pharmacies; No. = number; WHO = World Health Organization; % = percentage

(LPGs). Bearing in mind the low total sample size (n = 6) in Central province, 16.67% indicates medicines being available in one pharmacy only.

## Prices

The majority of medicines was sold at markedly higher prices than their international reference prices in all provinces (Table 3; values for Central province only available for LPGs). In total, 9 LPG antihypertensives had MPRs <1, indicating that Zambian patient prices for these are cheaper than external reference standards. An additional 4 generic diabetes and 20 hypertension medicines had acceptable prices (MPRs ≤2.5). The remainder had excessive MPRs above 2.5. Large variations in MPRs were observed across pharmacies (illustrated by the IQRs), individual medicines and between medicine forms. MPRs of OBPs were consistently, often multiple times, higher than that of their generic equivalents. The biggest differences were found for Amlodipine (5/10mg) and Glibenclamide, in which cases MPRs of OBPs were 28.39, 18.94 and 13.34 times greater than that of their generic form. Median unit prices can be found in S1 Table.

Fig 1 illustrates these variations for medicines found as both OBPs and LPGs in at least 4 pharmacies in Lusaka province. Based on the findings, the saving ratios showed savings associated with opting for generics between 21.54% and 92.5% for antidiabetics and 47.87% to 96.47% for antihypertensives.

## Affordability

In comparison to the national absolute poverty line, the monthly median treatment cost was considered unaffordable for the entire spectrum of surveyed medicines in all provinces. No standard treatment course with any of the medicines cost less or equal than one day's wage (affordability threshold). If affordability instead was measured against mean per capita monthly household income, the mean affordable proportion increased from 0% to 31% in Central and to 24% in Copperbelt and Lusaka province. Treatment with a number of LPGs lay within an affordable range in each province (Table 4). By contrast, OBPs were still unaffordable at the average provincial income levels with STC expenditures well above the threshold. The same applies to all insulins, requiring 3.29 to 7.20 days' salaries to purchase a 30-day STC.

Fig 2 illustrates the availability-affordability analysis in Lusaka province (using the mean per capita household income as base for affordability estimates). The x-axis represents the number of days' wages required to purchase an STC, the y-axis shows availability levels. Plotting the defined affordability threshold and the ≥80% availability target divides the plane into 4 quadrants: north-east (high availability/low affordability), south-east (low availability/low

**Table 2. Availability (in percentage) of surveyed antidiabetics and antihypertensives.**

| Province | Central | | Copperbelt | | Lusaka | |
|---|---|---|---|---|---|---|
| Medicine | OBP | LPG | OBP | LPG | OBP | LPG |
| Glibenclamide 5mg | 16.67 | 100 | 21.43 | 96.43 | 30.77 | 93.85 |
| Gliclazide 80mg | - | 50 | - | 32.14 | - | 38.46 |
| Glimepiride 2mg | 0 | 83.34 | 17.86 | 42.86 | 18.46 | 49.23 |
| Insulin 30/70 soluble/isophane | 16.67 | 33.34 | 39.29 | 10.71 | 29.23 | 9.23 |
| Insulin Intermediate-acting | 0 | 16.67 | 46.43 | 17.86 | 38.46 | 7.69 |
| Insulin Short-acting | 16.67 | 16.67 | 42.86 | 10.71 | 40 | 4.62 |
| Metformin 500mg | 0 | 100 | 7.14 | 100 | 9.23 | 100 |
| Mean availability all surveyed antidiabetics | 8.33 | 57.14 | 29.17 | 33.53 | 27.69 | 43.30 |
| | 34.62 | | 36.09 | | 37.37 | |
| Amlodipine 5mg | 0 | 100 | 7.14 | 96.43 | 13.85 | 92.31 |
| Amlodipine 10mg | 0 | 83.34 | 7.14 | 89.29 | 6.15 | 89.23 |
| Atenolol 50mg | 0 | 100 | 10.71 | 100 | 21.54 | 93.85 |
| Bendroflumethiazide 5mg | - | 16.67 | - | 50 | - | 38.46 |
| Bisoprolol 5mg | - | 50 | - | 57.14 | 4.62* | 66.15 |
| Captopril 25mg | - | 50 | - | 39.29 | - | 40 |
| Carvedilol 6.25mg | - | 33.34 | 7.14* | 42.86 | - | 56.92 |
| Carvedilol 25mg | - | 0 | 7.14* | 0 | 6.15* | 18.46 |
| Enalapril 5mg | - | 100 | - | 100 | - | 95.38 |
| Enalapril 10mg | - | 100 | - | 100 | - | 95.38 |
| Enalapril 20mg | - | 66.67 | - | 100 | - | 83.08 |
| Hydralazine 25mg | - | 0 | - | 21.43 | - | 21.54 |
| Hydralazine Injection | - | 0 | - | 3.57 | - | 6.15 |
| Hydrochlorothiazide 25mg | - | 66.67 | - | 71.43 | - | 52.31 |
| Hydrochlorothiazide 50mg | - | 0 | - | 17.86 | - | 30.78 |
| Lisinopril 5mg | 0 | 0 | 17.86 | 42.86 | 15.38 | 40 |
| Lisinopril 10mg | 0 | 0 | 17.86 | 39.29 | 16.92 | 41.54 |
| Lisinopril 20mg | 0 | 0 | 14.29 | 10.71 | 16.92 | 18.46 |
| Losartan 50mg | - | 100 | - | 89.29 | - | 83.08 |
| Metoprolol 100mg | - | 0 | 7.14* | 0 | 4.62* | 6.15 |
| Nifedipine 10mg | - | 83.34 | - | 89.29 | 1.54* | 80 |
| Nifedipine 20mg SR | - | 100 | - | 100 | - | 95.38 |
| Propranolol 10mg | 0 | 16.67 | 10.71 | 0 | 13.85 | 10.77 |
| Propranolol 40mg | 0 | 50 | 32.14 | 71.43 | 18.46 | 67.69 |
| Verapamil 40mg | - | 16.67 | - | 21.43 | - | 16.92 |
| Mean availability all surveyed antihypertensives | 0 | 46.67 | 14.73 | 54 | 15.38 | 53.60 |
| | 35.35 | | 37.23 | | 44.34 | |

LPG = lowest-priced generic; mg = milligram; OBP = original brand product; SR = sustained release.

-: Not surveyed original brand products due to lack of official market registration in Zambia

*: Original brand product without official market registration in Zambia that was found in surveyed pharmacies.

affordability), south-west (low availability/high affordability) and north-west (high availability/high affordability). 7 LPGs were located in the north-west quadrant and met both targets, including Glibenclamide for diabetes and Amlodipine (5/10mg), Atenolol, Enalapril (10/20mg) and Nifedipine (20mg SR) for hypertension. The majority of medicines for both NCDs (n = 26) was located in the south-east quadrant displaying low availability <80% and low

**Table 3. Median price ratios (interquartile ranges) of surveyed medicines.**

| Province | Central | | Copperbelt | | | | Lusaka | | | |
|---|---|---|---|---|---|---|---|---|---|---|
| Medicine form | LPG | | OBP | | LPG | | OBP | | LPG | |
| | MPR | (IQR) | MPR | (IQR) | MPR | (IQR) | MPR | (IQR) | MPR | (IQR) |
| Glibenclamide 5mg | 4.83 | (4.31–6.39) | 55.23 | (48.22–69.04) | 4.14 | (3.97–5.87) | 49.02 | (41.42–62.14) | 5.52 | (4.14–6.90) |
| Gliclazide 80mg | - | - | - | - | 2.26 | (1.60–2.61) | - | - | 1.62 | (1.54–2.42) |
| Glimepiride 2mg | 3.94 | (2.30–3.94) | 15.41 | (9.76–28.94) | 6.30 | (4.48–7.23) | 16.99 | (12.74–26.32) | 2.95 | (2.30–3.98) |
| Insulin 30/70 | - | - | 28.82 | (26.93–31.70) | - | - | 21.41 | (19.76–23.06) | 16.80 | (15.65–18.12) |
| Insulin Intermediate | - | - | 24.79 | (21.89–27.98) | 11.33 | (10.27–11.33) | 18.42 | (17.00–21.22) | 11.55 | (10.59–12.15) |
| Insulin Short-acting | - | - | 22.93 | (19.94–26.58) | - | - | 17.28 | (15.95–19.94) | - | - |
| Metformin 500mg | 2.23 | (1.64–2.62) | - | - | 2.1 | (1.57–2.62) | 10.49 | (10.49–10.49) | 2.62 | (1.57–2.62) |
| Amlodipine 5mg | 1.87 | (1.56–2.37) | - | - | 2.49 | (1.99–3.49) | 47.16 | (39.02–59.78) | 2.49 | (1.99–2.99) |
| Amlodipine 10mg | 1.38 | (1.27–2.12) | - | - | 1.59 | (1.06–1.70) | 36.06 | (31.91–36.06) | 1.27 | (1.06–1.70) |
| Atenolol 50mg | 2.76 | (2.30–3.49) | 42.03 | (39.40–42.03) | 2.94 | (2.21–3.68) | 39.14 | (29.42–42.03) | 3.68 | (2.63–4.41) |
| Bendroflum. 5mg | - | - | - | - | 7.36 | (6.74–7.60) | - | - | 7.12 | (5.89–7.60) |
| Bisoprolol 5mg | - | - | - | - | 2.11 | (1.98–2.47) | - | - | 1.47 | (1.25–1.87) |
| Captopril 25mg | - | - | - | - | 1.60 | (1.48–1.68) | - | - | 1.60 | (0.96–1.92) |
| Carvedilol 6.25mg | - | - | - | - | 4.10 | (3.56–4.71) | - | - | 3.37 | (2.58–3.67) |
| Carvedilol 25mg | - | - | - | - | - | - | 19.72 | (18.38–19.71) | 4.83 | (4.77–6.73) |
| Enalapril 5mg | 2.84 | (2.36–3.31) | - | - | 3.41 | (2.27–3.78) | - | - | 3.78 | (2.27–3.78) |
| Enalapril 10mg | 0.85 | (0.67–1.29) | - | - | 0.89 | (0.71–1.25) | - | - | 0.89 | (0.89–1.22) |
| Enalapril 20mg | 4.14 | (4.14–5.52) | - | - | 4.14 | (3.45–7.77) | - | - | 4.14 | (3.45–6.82) |
| Hydralazine 25mg | - | - | - | - | 4.64 | (2.51–5.80) | - | - | 5.00 | (4.51–5.80) |
| Hydralazine Injection | - | - | - | - | - | - | - | - | 0.46 | (0.20–0.71) |
| HCT 25mg | 6.86 | (4.35–11.44) | - | - | 17.39 | (9.15–20.13) | - | - | 13.73 | (9.15–18.30) |
| HCT 50mg | - | - | - | - | 8.03 | (8.03–20.08) | - | - | 8.03 | (8.03–16.06) |
| Lisinopril 5mg | - | - | 6.79 | (6.62–7.60) | 1.83 | (1.31–2.11) | 5.86 | (4.29–7.45) | 1.41 | (1.12–1.53) |
| Lisinopril 10mg | - | - | 4.61 | (4.49–5.16) | 1.43 | (1.14–1.55) | 3.97 | (3.57–5.34) | 1.11 | (0.95–1.40) |
| Lisinopril 20mg | - | - | 20.57 | (17.49–23.65) | - | - | 15.12 | (14.48–22.26) | 7.88 | (4.58–10.13) |
| Losartan 50mg | 0.82 | (0.80–0.97) | - | - | 0.80 | (0.68–0.91) | - | - | 0.81 | (0.58–1.03) |
| Metoprolol 100mg | - | - | - | - | - | - | - | - | 6.56 | (1.92–13.09) |
| Nifedipine 10mg | 0.83 | (0.83–0.95) | - | - | 0.95 | (0.71–1.19) | - | - | 1.19 | (0.92–1.42) |
| Nifedipine 20mg SR | 1.86 | (1.69–2.28) | - | - | 1.69 | (1.35–1.69) | - | - | 1.86 | (1.69–2.36) |
| Propranolol 10mg | - | - | - | - | - | - | 5.39 | (4.70–6.47) | 2.70 | (1.23–3.27) |
| Propranolol 40mg | - | - | 45.63 | (38.78–57.03) | 3.99 | (3.42–5.70) | 35.36 | (25.95–57.03) | 5.70 | (3.42–6.48) |
| Verapamil 50mg | - | - | - | - | 6.28 | (3.94–11.12) | - | - | 2.05 | (1.74–8.83) |

Bendroflum. = Bendroflumethiazide; HCT = Hydrochlorothiazide; IQR = interquartile range; LPG = lowest-priced generic; mg = milligram; MPR = median price ration; OBP = original brand product; SR = sustained release.

affordability. OBPs were without exception located in this quadrant. The same seven generics met both targets in Copperbelt; only Amlodipine (5/10mg) was found available and affordable in Central province.

## Registration status

Another finding was the registration status of the medicine names and manufacturers recorded in pharmacies. As LPGs were not prespecified, the actual products stocked varied greatly across pharmacies. Between 2 and 10 generic brands were found per medicine. Of these, 71.04% (Central), 60.85% (Copperbelt) and 54.69% (Lusaka) had ZAMRA market

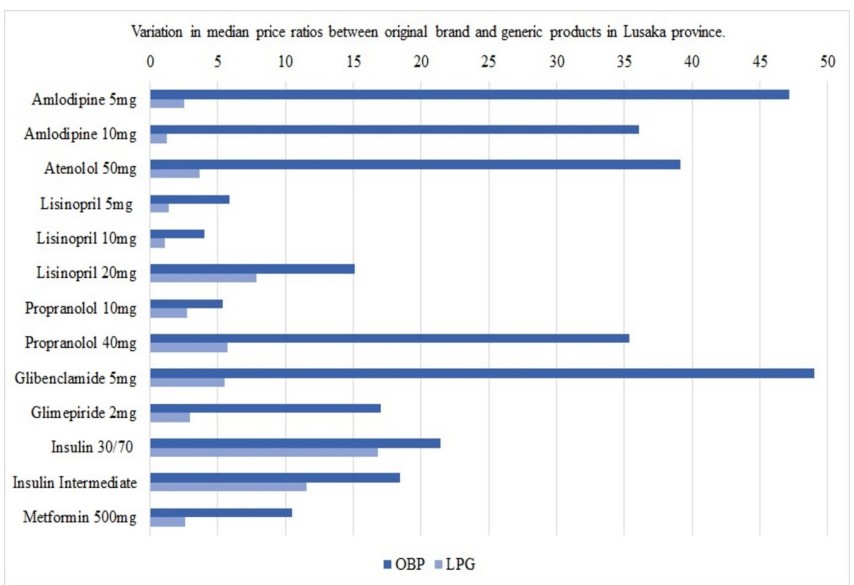

**Fig 1. Variations in median price ratios between original brand and generic products in Lusaka province.**
LPG = lowest-priced generic; mg = milligram; OBP = original brand product.

authorisation. No single generic product found in any pharmacy was officially registered for several LPGs, such as Bendroflumethiazide, Lisinopril (all strengths) and Verapamil.

## Discussion

To our knowledge, this is the first study investigating availability, prices and affordability of antidiabetic and antihypertensive medicines in private sector pharmacies in Zambia. The results showed that the majority of surveyed medicines was inadequately available (<80%),

**Table 4. Lowest-priced generics considered affordable (using mean per capita income).**

|  | Province | | |
| --- | --- | --- | --- |
|  | **Central** | **Copperbelt** | **Lusaka** |
| **National absolute poverty line level monthly income** | $13.15 ($0.63/day) | | |
| **Mean per capita monthly household income** | $24.10 ($1.15/day) | $46.25 ($2.20/day) | $48.85 ($2.33/day) |
| **Medicine** | | | |
| Glibenclamide 5mg | Not affordable (1.53) | 0.68 | 0.86 |
| Gliclazide 80mg | - | Not affordable (1.20) | 0.81 |
| Amlodipine 5mg | 0.82 | 0.57 | 0.54 |
| Amlodipine 10mg | 0.71 | 0.43 | 0.32 |
| Atenolol 50mg | Not affordable (1.23) | 0.68 | 0.81 |
| Bendroflumethiazide 5mg | - | 0.79 | 0.72 |
| Enalapril 10mg | Not affordable (1.04) | 0.57 | 0.54 |
| Enalapril 20mg | 0.66 | 0.34 | 0.32 |
| Hydrochlorothiazide 25mg | 0.82 | Not affordable (1.08) | 0.81 |
| Hydrochlorothiazide 50mg | - | 0.28 | 0.27 |
| Nifedipine 20mg SR | Not affordable (1.80) | 0.85 | 0.81 |

mg = milligram; SR = sustained release; $ = US-Dollar.

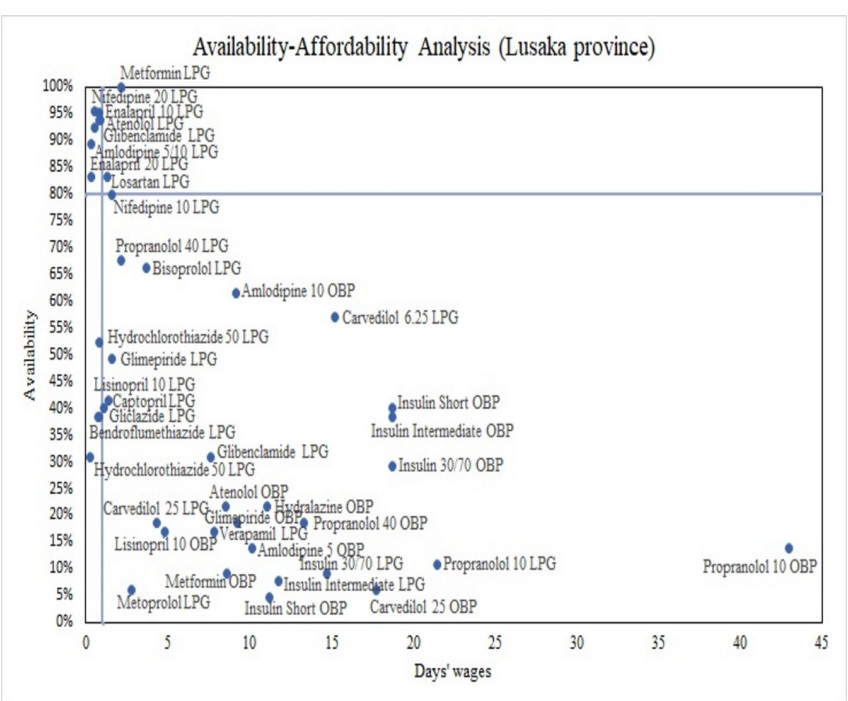

**Fig 2. Availability-affordability analysis of surveyed medicines in Lusaka province.** LPG = Lowest-priced generic; OBP = original brand product.

most prices were high in an international comparison and treatment with these medicines was largely unaffordable against the set affordability thresholds.

## Availability

Mean availability of all medicines across pharmacies and the two surveyed baskets of medicines was low, which reflects the inadequate availability (<80%) of the majority of individual medicines. Nevertheless, it is important not to overrate low availability as pharmacies may stock different strengths of the medicines and the applied cross-sectional design does not capture stock fluctuations over time. Individual medicine availability should further be considered in the light of availability of other medicines within the same therapeutic class that would allow for therapeutic substitution [40] (e.g. Lisinopril/Enalapril). As no substitution possibilities exist for insulin, access is vital for treatment success and ultimately life-saving [3] and its low availability hence striking. In 2005, Beran et al. found insulin available at all 13 public hospitals and 42% of health centres surveyed across three Zambian provinces [41], yet in 2016, Kalungia et al. found low-level availability of 22.2% to 37.8% at 15 surveyed public facilities in Lusaka [12]. Moreover, nearly all medicines were more widely available as generics compared to OBPs, which was reflected in the results of similar studies [7–10,33,42–49]. In contrast, the higher availability of original compared to biosimilar insulin (Copperbelt and Lusaka) was remarkable. This could be explained by the lack of patient acceptance due to perceived lower effectiveness of biosimilar insulin [50] and the limited competition on the global insulin market. Novo Nordisk, Eli Lilly and Sanofi Aventis dominate and control about 96% of the market volume and 99% of its value [51]. Further, they have direct distribution agreements with countries [52], which is reflected by the fact that insulins included in the ZEML are partly listed by Novo Nordisk's product brand names rather than their international non-proprietary names [24].

In similar studies, with few exceptions, Metformin and Glibenclamide were also found the most widely available medicines [9,32,33,43,44,47,49]. Ewen et al. reported higher private sector availability (66%) for a nearly identical basket of antidiabetics in 12 LMICs [10], thus indicating improvement potential in the Zambian provinces. Two large-scale reviews reported estimates for generic Atenolol, Captopril, Hydrochlorothiazide, Losartan and Nifedipine [8,32]; availability of these medicines was markedly higher in the present study, which, given generics' generally lower prices, can be evaluated positively.

The poor availability is likely to be multifactorial. Apart from those mentioned, further possible explanations might be cash-flow constraints [2], inadequate local manufacturing capacities and resulting import dependencies [1,6], poor demand forecasting due to inadequate HMIS in many surveyed pharmacies and purchasing not being guided by the ZEML. Pharmacy managers further highlighted the varying delivery lead times based on location, size and working capital of pharmacies and further the close relationship between stock and demand (they predominantly stock fast-moving products as other medicines run a high risk of expiring). Ordering specific medicines on request or receiving stock in between regularly scheduled deliveries were indicated as possibilities–actual feasibility, however, is in turn dependent on the factors influencing delivery lead times and hardly possible for smaller or remotely located pharmacies. Further research is needed to document the significance of these findings. Similarly, pharmaceutical market mechanisms and business conditions in Zambia require further investigation, not least to identify ways of increasing availability to clearly needed antidiabetics and antihypertensives.

## Prices

On the whole, median price ratios of the surveyed medicines were largely found above the stipulated $\leq 2.5$ threshold. Values of OBPs were consistently higher than of generic equivalents, which is underpinned by the high saving ratios between 21.54% and 96.47%. The results of similar surveys in LMICs corroborate these findings [8,9,46–49,10,13,32,41–45]. The variability in MPRs between individual medicines has also become apparent in similar surveys, albeit with considerably different MPRs even for the same medicines. For instance, MPRs ranged from 2.8 [42] to 44.31 [8] for Atenolol, from 0.15 [32] to 14.70 [8] for Captopril and lay between 3.42 [9] and 37.28 for Glibenclamide (LPGs) [13]. These differences are indicative of the various factors influencing MPRs, including country-specific conditions (medicine prices, procurement systems, currency strength), IRPs and the year when studies were performed.

MPRs of insulin were further noticeable, since even biosimilars were priced at 11.33 to 16.80 times their IRP. This coincides with the findings of other studies [53,54]. Within Novo Nordisk's "Access to Insulin Commitment" initiative, it supplies insulin to LMICs' governments at a ceiling price of US$4 (vial) [55]. This falls substantially below median prices found in the surveyed provinces (US$10.87 to US$14.42) and thus confirms the findings of 85–125% higher insulin prices from private wholesalers [41]. At present prices, yearly treatment would cost 2.04 to 2.74 times the total health expenditure per capita in Zambia of US$69.4 (2015) [56] and 9.3% to 12.3% of annual GDP per capita of US$1513.3 (2017) [57]. Pricing data reported echo the high prices found, albeit with great variations. In the WHO African region, HAI reported that prices for short-acting insulin ranged from US$3 (Senegal) to US$47 (Namibia) [58].

Several reasons might explain the high MPRs. Firstly, excessive fluctuations in foreign currency exchange rates and the depreciation of the ZMW-US$ parity have led to price instability and increases in prices for import goods [59]. As the bulk of medicines is imported and procured in US$ [1,6], it can be expected that such externalities have forced prices to go up.

Further, this study focused on patient prices and procurement prices and charges (e.g. mark-ups, taxes, distribution costs) accruing throughout the supply chain were not considered. These were shown in other studies to account for price differences and increases in studies including price component analyses, with retail mark-ups being the highest contributor [8,9,13,32,34,46,48,49]. As medicines are exempted from import tariffs and VAT in Zambia [60,61], present study results similarly suggest substantial mark-ups for the surveyed medicines. Moreover, there are currently no legal or regulatory provisions regarding medicine pricing in the private sector [62]. For instance, mark-ups are neither reduced nor regulated [62] and prices thus, at best, determined by market forces. The resulting financial scope and inconsistencies in mark-ups could hence provide further explanations for the observed MPR variations. In addition, there is little competition in the supply of antihypertensive and antidiabetic medicines; the largest seller accounts for 91.7% and 100% of all sales, which has a direct bearing on procurement and patient prices of these medicines [63]. Exact causes for the high prices found, however, need to be ascertained through a price components study and additional explorative studies.

## Affordability

Medicine affordability is not a straightforward concept and lacks a clear definition [10,64]. However, the proposed measure in the WHO/HAI methods (number of days' wages to purchase a standard treatment cost) has been widely recognised as it clearly and unambiguously shows the financial burden for those having to purchase their medicines individually [10,18]. In the present study, affordability of surveyed antidiabetics and antihypertensives against the defined threshold (STC ≤ one day's wage) was very low; no medicine was considered affordable against the national absolute poverty line and only 4 (Central), 9 (Copperbelt) and 11 LPGs (Lusaka) had treatment costs below the threshold when using provinces' mean per capita monthly household incomes as measure. Hence, these results indicate that hypertension and diabetes place a high economic burden on individuals and households at these or lower income levels.

A 2009 study found Glibenclamide, Captopril, Hydrochlorothiazide and Nifedipine affordable in the private sector in China using the Chinese absolute poverty line (US$0.4) [34] as base for defining affordability, thus contrasting the present findings. Most other relevant surveys based the affordability assessment on the salary of the lowest paid unskilled government worker, which limits comparability of results from various studies. However, as that salary generally is higher than the affordability thresholds used in this study, one can assume that medicines considered unaffordable in those other studies would also be classified as that in the present analysis. Firstly, treatment costs with OBPs in general [8,10,32–34,42,43,45,47,65] and with any original or biosimilar insulin in particular [10,33,42,43,45–49,65] were also found unaffordable against the threshold. Further coinciding with the present findings, Glibenclamide and Hydrochlorothiazide were found to be the most affordable medicines for treating diabetes and hypertension, respectively [32–34,42–45,47]. Results for the remaining medicines differed across studies. For instance, as was the case in Copperbelt and Lusaka province, Atenolol, Enalapril and Nifedipine were affordable in several studies [32,34,44] and exceeded the threshold in others [42,43,45]. This shows the high context-sensitivity of affordability analyses and the dependency of results on factors such as medicine prices and income levels.

Additional aspects need to be considered in relation to affordability. First, the absolute poverty line was used instead of the extreme (food) poverty line [19]. However, a large number of the provinces' residents have incomes below the absolute poverty line [19], making medicines even more unaffordable for them. Income distribution should also be considered when

interpreting affordability based on provincial average incomes. The gini coefficients of around 0.50 indicate high economic inequalities in all provinces [19]–even medicines considered affordable are thus still beyond the reach of many patients in these provinces. Further, despite the threshold, it is difficult to accurately assess affordability, since it strongly depends on each patient's situation. For instance, defined daily dosages mask individual dosage requirements and intake of only one medicine was assumed in the analysis. Assessments were based on medicine costs alone and disregarded additional expenses, such as cost of supplies for medicine administration and travel costs. Including such factors could change affordability results for the worse. Moreover, the low average quantity sold by most pharmacies further emerged as crucial. Medicines were almost exclusively sold in strips (10 tablets) or as single tablets. Pharmacists' explanation to this was that the limited financial means of many customers prevented them from buying full STCs. Instead, they need to visit the pharmacies several times a month (or even daily) in order to obtain the required medication in smaller quantities at, hence, lower costs at one time. In light of mentioned travel costs and as bulk-breaking has been shown to entail higher mark-ups and prices [2], the purchase of smaller quantities could further compromise affordability. Additionally, it illustrates the close link between affordability and availability: if patients cannot afford to build up adequate private medicine reserves, unavailability can lead to erratic treatment and people having to forego care. As with availability, the significance of this information needs to be documented by further research. Further, it is vital to pay due attention to long-term affordability, which has been shown a main cause of poor adherence to long-term treatment for NCDs and impoverishment [66–68]; future research should include such analyses.

## Registration status

The nature of medicines makes it impossible for patients to distinguish levels of quality and efficacy [69]. Consequently, the lack of registration detected may have serious effects on individual and public health. Quality, efficacy and safety of unregistered medicines are not adequately tested and ensured, hence increasing the possibility of substandard, falsely labelled, and falsified products entering the market [69]. WHO estimates that one in ten medicines in LMICs are falsified or substandard [70], a fact that also raises concerns for Zambia.

## Study limitations

As provinces were purposively selected, the possibility of any selection bias of provinces introduced cannot be entirely ruled out. A larger study surveying the remaining 7 provinces should be conducted to allow for conclusive statements on availability, prices and affordability of antidiabetics and antihypertensives in private pharmacies in Zambia. Data on all variables were collected at a single time point for each pharmacy, which might not give a correct picture of the current or long-term availability and/or prices. Future studies should examine these variables to allow for an assessment of the commodity security of antidiabetic and antihypertensive medicines. Further, the list of surveyed medicines was not exhaustive. It did not account for alternate strengths or dosage forms and medicines found in many pharmacies (e.g. Telmisartan and fixed-dose combinations including Metformin-Glimepiride, Hydrochlorothiazide-Losartan) were excluded (no IRPs available in the MSH price set). Future research would benefit from recording availability and price information of such medicines, even if MPRs cannot be calculated. Moreover, many pharmacies had no price lists or IT-systems in place to extract medicine prices, making it necessary for the researcher to rely on information provided by pharmacy staff. To minimize bias and the possibility of incorrect information given, data were recorded with either pharmacists or pharmacy technicians and one can thus assume that valid

and reliable data were obtained. Moreover, price assessments relied on median price ratios, the accuracy of which depends on the reliability of IRPs published by MSH. If these median values are based on few available data and thus potentially skewed by high/low supplier prices, this biases MPRs and makes the Zambian prices appear more/less favourable in an international comparison. Lastly, the qualitative information included in the discussion obtained in from participating pharmacy staff warrants further examination in future qualitative studies.

## Conclusion

This study provided evidence on availability, prices and affordability of antidiabetic and antihypertensive medicines in private pharmacies in Central, Copperbelt and Lusaka province of Zambia. Diabetes and hypertension patients need a reliable supply of affordable medicines to avoid preventable morbidity and mortality. However, study results showed that the majority of surveyed antidiabetics and antihypertensives were insufficiently available and that they were largely unaffordable against the defined thresholds. The next steps now needed are to assess the exact causes for these results and to conduct similar surveys in the other Zambian provinces as well as in the public sector to enable conclusive statements about access to antidiabetic and antihypertensive medicines for the health sector as a whole. On this basis, policy measures and strategies should then be devised and implemented at various levels of the Zambian healthcare system and more generally the pharmaceutical market.

## Supporting information

**S1 Table. Median unit prices (in 2019 US-Dollars) of surveyed antidiabetics and antihypertensives.**
(DOCX)

## Acknowledgments

We would like to thank all pharmacy staff working at the 99 surveyed pharmacies for their effort and assistance with data collection. Selected references and recommendations given within the scope of this study are those of the authors alone and do not necessarily represent the views of CHAI and or any endorsement on their part.

## Author Contributions

**Conceptualization:** Lindsey Hehman, Jesper Sundewall.

**Formal analysis:** Andrea Hannah Kaiser.

**Investigation:** Andrea Hannah Kaiser.

**Methodology:** Andrea Hannah Kaiser.

**Supervision:** Jesper Sundewall.

**Validation:** Birger Carl Forsberg.

**Visualization:** Andrea Hannah Kaiser, Warren Mukelabai Simangolwa.

**Writing – original draft:** Andrea Hannah Kaiser.

**Writing – review & editing:** Andrea Hannah Kaiser, Lindsey Hehman, Birger Carl Forsberg, Warren Mukelabai Simangolwa, Jesper Sundewall.

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
