## [Decision Letter · Decision Letter 0]

16 Sep 2019

PONE-D-19-20413

Availability, prices and affordability of essential medicines for treatment of diabetes and hypertension in the private sector in Zambia.

PLOS ONE

Dear Dr. Forsberg,

Thank you for submitting your manuscript to PLOS ONE. After careful consideration, we feel that it has merit but does not fully meet PLOS ONE’s publication criteria as it currently stands. Therefore, we invite you to submit a revised version of the manuscript that addresses the points raised during the review process.

We would appreciate receiving your revised manuscript by Oct 31 2019 11:59PM. To enhance the reproducibility of your results, we recommend that if applicable you deposit your laboratory protocols in protocols.io, where a protocol can be assigned its own identifier (DOI) such that it can be cited independently in the future. For instructions see: http://journals.plos.org/plosone/s/submission-guidelines#loc-laboratory-protocols

We look forward to receiving your revised manuscript.

Kind regards,

Khin Thet Wai, MBBS, MPH, MA (Population & Family Planning Resear

Academic Editor

PLOS ONE

Journal Requirements:

Additional Editor Comments (if provided):

This paper used the WHO standardized methodology and could be able to bring out the policy-related evidence to improve access to antidiabetics and antihypertensive in the private sector at Zambia.

The significant difference median price ratios for LPG between the study sites requires an appropriate statistical test.

Apart from that authors should respond thoroughly to the modest request of reviewers to deal with sampling issues, some results and the discussion part to improve the writeup.

Reviewers' comments:

Reviewer's Responses to Questions

**Comments to the Author**

1. Is the manuscript technically sound, and do the data support the conclusions?

Reviewer #1: Yes

Reviewer #2: Yes

2. Has the statistical analysis been performed appropriately and rigorously? 

Reviewer #1: Yes

Reviewer #2: Yes

3. Have the authors made all data underlying the findings in their manuscript fully available?

Reviewer #1: Yes

Reviewer #2: Yes

4. Is the manuscript presented in an intelligible fashion and written in standard English?

Reviewer #1: Yes

Reviewer #2: Yes

5. Review Comments to the Author

Reviewer #1: Line 31: "In line with its pledge to achieve Universal Health Coverage by 2030, the government..". The authors have to indicate the country as this is the first statement in the introduction.

Line 121-122: "To verify effectiveness of the data collection approach and accuracy of the tool, local pilot surveys were conducted at two pharmacies in Lusaka". Was this data included in the final analysis for this paper?

Line 197-199: "Of the 32 selected medicines, 12 had no OBP registered with ZAMRA. The survey drugs thus

198 comprised 14 OBPs (Diabetes: 6; Hypertension: 8) and 32 LPGs (Diabetes: 7; Hypertension:

199 25)." The number of OBP (12 + 14) does not match with LPG (32). Is there an explanation for this?

Line 240: "Table 3. Median price ratios (interquartile ranges) of surveyed drugs." Preference would be to include the results of Central province in the same table than text below it.

Line 259: "Affordability." The authors could have used the minimum wage for Zambia than the 'national absolute poverty line' as the study was conducted in major cities where majority of the individuals are employed in various sectors.

Reviewer #2: GENERAL COMMENTS:

The paper explored an important subject of availability, pricing and affordability of antidiabetic and antihypertensive medicines in the private sector in Zambia. Addressing treatment challenges for non-communicable diseases in low-middle income countries does require evidence to inform policy and practice interventions towards the universal goals of ensuring access to essential medicines. Though the paper was limited in scope and coverage in terms of study sites, a country-wide study would add value to the findings providing insight into the magnitude of the medicine availability for diabetes mellitus and hypertension treatment in Zambia’s private sector. Overall, the paper is relevant, well structured, has sufficient scholarly rigor, and has a logical flow of arguments, notwithstanding the limitations highlighted by authors.

SPECIFIC COMMENTS:

The title is a bit misleading. Reading through the paper, the scope of population coverage was rather limited. The paper only focussed on measuring availability, pricing and affordability in the private pharmacies in Zambia, which does not necessarily represent the entire scope of private sector providers of medicines. I note that the study did not cover the other private sector providers such as private hospitals and consulting room dispensaries. I assume these private players also do provide antidiabetic and antihypertensive medicines in the setting (Zambia)? Since the coverage of study sites was limited, I can suggest the authors revise and refocus their title to state “…availability, price and affordability in the private pharmacies in Zambia”

Line 4: It is good that the authors sampled the pharmacies from the list of registered (legal) outlets, however, among the 99 private pharmacies surveyed, the authors do not state the distribution of these pharmacies in the setting, whether and how many of these pharmacies were located in urban versus rural areas of Zambia? Did availability, pricing and affordability vary by location of the pharmacy in the setting?

Line 7: Regarding reasons why “international reference pricing” was used as a measure of affordability, did the authors consider difference in pricing considerations of originator brand products, local brands, versus ‘generic’ products, including cost variations between European versus Asian manufactured products on the market? Was there a local costing structure for medicines in Zambia? I will appreciate a comment on that.

Line 13: In the results section, a bit more clarity required when stating the findings. Some missing words and stylistic aspects in the opening sentence e.g. opening a sentence with a numeral?

Line 14: Authors state: “availability levels for the remainder …..”, but what is the remainder in this case? How many medicines were the remainder? The sentence is not clear. Consider using absolute measures.

Line 31: Which “government” are authors referring to? The opening paragraph (Line 31 to 35) needs to be recast and context provided.

Authors seem to be using the terms “drug” and “medicine” interchangeably? I highly doubt that the two terms are synonyms. I can suggest authors maintain to use one term consistently throughout the paper.

Line 38: Did the authors select the three provinces conveniently or purposively? This is not clear and if so, why? If not, a statement should be provided how authors ensured selection bias of provinces was avoided.

Line 98 – 100: Was a sample size formula applied to select the 99 pharmacies from the total population of private pharmacies? If so, what were the assumptions forwarded in the computation of sample size (notwithstanding the methods developed by Measure Evaluation)?

Moreover, how was the site selection (private pharmacy) done? Was due consideration given to systematically or randomly select the pharmacies or otherwise? A clear statement on this will be appreciated.

Line 103 – 104: How did the authors arrive at 10% for adjusting sample size? Any explanation or justification for this? Why 10% and not say 15% or 20%? What was the considered assumption for non-response rate?

Line 108: Among the medicines that consisted the 7 antidiabetic and 25 antihypertensive medicines, it is not clear whether authors were only interested in the single formulation medicines, or even co-formulated products?

It is a shame that if indeed the medicine was not available (perhaps stocked out) at the respective pharmacies on spot survey at time of data collection, the data collection tool would have also done well to collect data on whether the medicine was ever stocked, the number of stock-out days in a period specified, and so on. This finding could potentially inform whether there were underlying issues with medicine demand or supply challenges in Zambia. I feel readers of this paper would also be interested to know this information.

Line 145 - 147: The measure of availability that was used by Authors was rather unreliable, despite the method being widely applied by previous studies. There is a difference in ‘on the spot availability’ and ‘consistent availability’. It appears to me the authors measured ‘on the spot availability or presence’ of the medicines on spot survey. Assessing availability of medicines, in the context of pharmaceutical supply chain management principles, requires measuring commodity security i.e. consistency of availability to maintain steady flow of the product to majority consumers at all times. A medicine must be in stock in sufficient quantities to meet demand for a specified period of time consistently to date (without stock out) for it to be deemed available. Was commodity security of supply a consideration or on the spot availability of the product on the shelf?

Line 167: “STC” should be written in full when first used, then abbreviated thereafter.

Line 177: The subheading sounds repetitive of the two previous subheadings used. I feel this section from line 177 to 182 is not adding any value to the paper. If the information is needed, authors can take move the sentences to other sections.

Line 223: Am wondering if private pharmacies in the setting not having available some of the medicines surveyed is a surprising finding. Am not surprised by this finding since ideally, most if not all these antidiabetic and antihypertensive medicines are supplied on prescription-only basis to clients accessing private healthcare. In the private (for-profit) sector, commodity availability is often driven by demand (from prescriptions being received at the private pharmacy). Therefore, we wouldn’t expect a particular medicine that is not frequently prescribed to be so available in all the private pharmacies, more especially if that medicine is not on the EML or national standard treatment guideline (STG) for diabetes or hypertension. In actual fact, to be prescribed that often the medicine has to be at least the choice specified by the national treatment guidelines thereof. Did the authors consider this possible cause negatively impacting on availability in the pharmacies? So to avoid distorting the availability picture, Authors would do well to focus on those medicines that actually have marketing authorisation and feature on the National Formulary and/or STG rather than the WHO lists as reference of what is actually recommended to be prescribed for diabetes and hypertension treatment in Zambia. A discussion point perhaps?

The findings show that even some medicines without official marketing authorisation were actually selling at a price in the private pharmacies in Zambia. If some of the pharmacies were actually stocking and selling antidiabetic and antihypertensive medicines that had no official marketing authorisation from the regulator, did that imply there was illegal sale and distribution of some antidiabetic and antihypertensive medicines in private pharmacies in Zambia? Was that a weakness in the regulatory aspects? Do you want to comment on that?

Line 270: Regarding pricing affordability for generic brands, I feel the findings in table 4 would be better interpreted when the actual average price paid per unit of sale is shown (whether per pack, number of tablets, or monthly supply quoted either in local or US dollar currency equivalent) and also the affordability index (using mean per capita income) indicated in the table 4.

Line 335, 427 and elsewhere: In the discussion, Authors bring out what appears to be some qualitative findings from informant responses (e.g. from Pharmacy managers about “varying delivery times” and so on). If at all there was a qualitative component in the study, why were the qualitative findings/results? Did the data collection also involve qualitative data? Were there any qualitative findings in the study? This needs to be clarified from the onset.

Line 447 – 460: Authors bring out a number of important limitations of the study. They can also do well to mention how these limitations were addressed or can be mitigated in future studies.

Line 462 – 471: The conclusion needs to be recast. Line 462 to 464 are not adding value to the section. Rather than general statements, authors will do well to address the research objective(s) by mentioning the overarching findings of the study.

6. PLOS authors have the option to publish the peer review history of their article (what does this mean?). If published, this will include your full peer review and any attached files.

Reviewer #1: Yes: Felix Khuluza, PhD

Reviewer #2: Yes: Aubrey Chichonyi Kalungia

---

## [Author Response · Author response to Decision Letter 0]

22 Oct 2019

We would like to thank the reviewers for their valuable comments and feedback. In the following, we respond to their comments. Please see the uploaded word document "Response to Reviewers", which contains the same information in tabular format for the sake of improved clarity.

Reviewer 1

31 “Republic of Zambia” has been added in LL33-34 to make it clear which country we are referring to.

121-122 A brief clarification that this data was not included in the final analysis for this paper has been added in LL153.

197-199 A brief statement has been included in LL236-237 explaining that the drugs without official ZAMRA registration were excluded from the survey.

240 We agree with the reviewer as regards the inclusion of the Central Province values in Table 3. The layout has been adjusted accordingly to allow for an additional column. Only values for LPGs have been included as no MPRs could be calculated for any OBP antihypertensive/antidiabetic medicine in Central province (none was available in at least 4 pharmacies).

259 We agree that using the minimum wage would certainly have given valuable answers as regards medicine affordability for those having incomes at or around this threshold. However, conducting the analyses with both Zambia’s absolute poverty line and the mean per capita monthly household income in the three provinces was a deliberate choice to represent both the poor and the social average. Firstly, especially since a considerable share of the provinces’ population had a disposable income below the national absolute poverty line (56.2% in Central, 30.8% in Copperbelt and 20.2% in Lusaka province – the national absolute poverty line lay at $13.15 per month at the time of the survey), using this measure seemed appropriate to represent this population share. Secondly, the average monthly incomes ($24.10 in Central, $46.25 in Copperbelt and $48.85 in Lusaka per month at the time of the survey) are considerably lower than the minimum wage mandated by law in Zambia (ZKW 1050 per month, equaling $87.76 using the exchange rate applied in the survey). This shows the discrepancy between the level of minimum salary applying to the share of the population with formal sector jobs and people’s actual income, most of which earn their incomes in the informal sector. On these grounds, we propose to keep the actual (mean) per capita household income as a second affordability measure.

Reviewer 2

Title The title and the objective have been revised accordingly.

4 A statement has been added about the urbanization status of the three chosen provinces (LL97-98). An additional sentence has been added in the results (LL242-245) to elaborate on the distribution of the pharmacies in the study area. As the pharmacies were invariably located in urban parts of Zambia, differences in availability, prices or affordability could not be assessed.

7 Using IRPs was part of the WHO/HAI methodology and the purpose is to make different studies from different countries comparable and highlight high medicine prices as a basis for potential policy interventions. While designing the methodology, consideration was given to potentially calculate such price differences. However, the actual data collection showed that the vast majority of medicines originated from Asia, particularly India, and only a few medicines, particularly antihypertensives, were imported from Europe. It hence did not seem that considering cost variations between the European/Asian manufactured products would be very informative. Furthermore, no product found in any pharmacy was manufactured in Zambia, hence making comparisons to local brands impossible. LL425-429 refer to the costing structure for medicines in Zambia. “Moreover, there are currently no legal or regulatory provisions regarding medicine pricing in the private sector. For instance, mark-ups are neither reduced nor regulated and prices thus, at best, determined by market forces. The resulting financial scope and inconsistencies in mark-ups could hence provide further explanations for the observed MPR variations.” Hence, there is no specific costing structure for medicines in the Zambian private retail pharmacy sector, but prices are determined by the pharmacists and the competition they are facing.

13 The results section has been slightly rephrased to improve the mentioned stylistic aspects.

14 This sentence has been rephrased to provide more clarity.

31 “Republic of Zambia” has been added in L34 to make it clear which country and government we are referring to.

 We agree with the reviewer and all “drug” and “drugs” has been replaced with “medicine” and “medicines”, respectively in order to use one term consistently throughout the paper.

38 The section “survey scope” has been revised (LL91-110). In addition, a statement has been added to the limitations section to acknowledge the possibility of selection bias.

98-100 More detail has been provided on the pharmacy selection as well as the formulas used to determine the sample sizes (LL116-123). From the methods developed by Measure Evaluation, Formula 1 (see word document) was used to determine the sample size. Precision requirements were set at p � 0.2p at the 90% confidence level (t = 1.64), resulting in a relative error of 20% (relative variance V2 = 0.22). No data were available from previous surveys regarding sample design effects; as the ZAMRA list was not clustered, it was assumed low (f = 1.2). The anticipated proportion of pharmacies with the attribute of interest was similarly unknown; in line with the specifications of the methods used, p = 0.5 and hence q = 1–p = 0.5 were used, in which case the sample size is maximized. Inserting these values resulted in a sample size of n = 115 for each province. As the total population of pharmacies was predetermined and relatively small in both provinces (N = 149 in Lusaka and N = 36 in Copperbelt province), the resulting n was greater than 5% of the total. In such cases, sample size is to be adjusted with the Finite Population Correction factor (Formula 2 in word document). After correction, resulting sample sizes were nnew = 65 for Lusaka and nnew = 28 for Copperbelt province. In addition, a brief statement has been provided in LL123-126 on how the random selection was done.

103-104 Absent any other prior information, WHO’s Service Availability and Readiness Assessment (SARA) recommends increasing the sample size by at least 10% to take into account non-response, and 10% is similarly used as an example in the consulted sampling manual published by Measure Evaluation. On this basis and in the absence of any other information on private sector non-response rates in Zambia, 10% was chosen.

108 In principle, both single-ingredient and fixed-dose combinations were considered for inclusion in the survey. However, no fixed-dose combination product fulfilled both inclusion criteria, i.e. had an IRP registered in the MSH Drug Price Indicator Guide and was registered with ZAMRA. Hence, only single formulation medicines were included in the final survey list. The lack of fixed-dose combination products in the survey list is further acknowledged in the study limitations. A statement has been added in LL156-157 (data collection) to alert the reader that data was only collected at a single time point for each pharmacy. This approach was chosen in keeping with the applied WHO/HAI methodology. In addition, while the suggested data collection approach would have been preferable, most pharmacies did not have a reliable information system keeping track of stock-out days, etc., hence rendering this approach very difficult and/or impossible. In addition, the pitfalls of the chosen data collection approach are further acknowledged in the study limitations (LL 502-506).

145-147 This comment seems directly related to the previous comment. As this study employed a descriptive, cross-sectional study design, commodity security was not considered but the focus was on actual availability on the shelf at the time of the survey. More information on availability is also provided in the reply to the comment on L223.

167 This abbreviation was introduced in L151 (Section: Selection of medicines).

177 The methods section on affordability has been merged with the availability/affordability section and the subheading for the latter was taken out (LL214-221). The same has been done in the results section (LL317-319).

223 We agree with the reviewer that the finding is not very surprising- despite the very low availability of many medicines being striking. We have addressed the reviewer’s comment in the discussion in the for instance the following (LL382ff): “Pharmacy managers further highlighted the varying delivery lead times based on location, size and working capital of pharmacies and further the close relationship between stock and demand (they predominantly stock fast-moving products as other drugs run a high risk of expiring). Ordering specific drugs on request or receiving stock in between regularly scheduled deliveries were indicated as possibilities – actual feasibility, however, is in turn dependent on the factors influencing delivery lead times and hardly possible for smaller or remotely located pharmacies.” In addition, as detailed in the medicine selection paragraph in the methods section, we briefly mentioned that the Zambia National Formulary and standard diabetes and hypertension treatment guidelines were consulted for the medicine selection in order not to miss medicines that are frequently prescribed but not included on either the WHO EML or the ZEML. The findings that unregistered medicines were sold in the surveyed pharmacies implies illegal sale and distribution to some extent. Pharmacies are not allowed to bulk purchase unregistered drugs and/or to regularly sell them. However, they are allowed to import a certain quantity of those drugs on request by specific patients. However, as many unregistered products were regularly found in many of the surveyed pharmacies and sometimes even the only products available, the assumption can be made that illegal sale and distribution was happening and that there are major regulatory weaknesses.

270 A table illustrating median unit prices for all antidiabetics and antihypertensives for all three provinces has been uploaded as a supplement and this has been pointed out to the reader (LL284-285). Moreover, two additional rows have been added to show both the level of the absolute national poverty line and the mean per capita household incomes per month and per day (in 2019 USD).

335, 427, etc. The data collection did not involve qualitative data collection per se, yet several pharmacy staff highlighted the included issues, which were recorded in the researcher’s field notes. The obtained information was verified in additional discussions with practicing pharmacists as well as officials at Medical Stores Limited and on this basis, we decided to include the information to inform the discussion section of the present paper. A brief statement has been added in the data collection section (LL172-175) with regards to the qualitative information included in the discussion. In addition, the importance of documenting the significance of this information by conducting future qualitative studies has been highlighted in the discussion (LL388-389; LL485-486). We have also included a statement in the limitations section to point out to the reader that the qualitative information warrants careful interpretation (LL521-522).

447-460 We agree with the reviewer that elaborating more on the limitations and highlighting possibilities to mitigate these for future studies is important. The limitation section has been rephrased accordingly and recommendations for future studies have been provided.

462-471 The conclusion has been rephrased and overarching findings of the study have been mentioned.

---

## [Decision Letter · Decision Letter 1]

21 Nov 2019

Availability, prices and affordability of essential medicines for treatment of diabetes and hypertension in private pharmacies in Zambia.

PONE-D-19-20413R1

Dear Dr. Forsberg,

We are pleased to inform you that your manuscript has been judged scientifically suitable for publication and will be formally accepted for publication once it complies with all outstanding technical requirements.

With kind regards,

Khin Thet Wai, MBBS, MPH, MA (Population & Family Planning Resear

Academic Editor

PLOS ONE

Additional Editor Comments (optional):

All comments of reviewers are fully addressed.

Reviewers' comments:

Reviewer's Responses to Questions

**Comments to the Author**

1. If the authors have adequately addressed your comments raised in a previous round of review and you feel that this manuscript is now acceptable for publication, you may indicate that here to bypass the “Comments to the Author” section, enter your conflict of interest statement in the “Confidential to Editor” section, and submit your "Accept" recommendation.

Reviewer #1: All comments have been addressed

2. Is the manuscript technically sound, and do the data support the conclusions?

Reviewer #1: Yes

3. Has the statistical analysis been performed appropriately and rigorously? 

Reviewer #1: Yes

4. Have the authors made all data underlying the findings in their manuscript fully available?

Reviewer #1: Yes

5. Is the manuscript presented in an intelligible fashion and written in standard English?

Reviewer #1: Yes

6. Review Comments to the Author

Reviewer #1: The authors have addressed all the comments. In addition, the authors have provided explanations where changes to my suggestions could not be taken on board.

7. PLOS authors have the option to publish the peer review history of their article (what does this mean?). If published, this will include your full peer review and any attached files.

Reviewer #1: Yes: Dr. Felix Khuluza

---

## [Editor Report · Acceptance letter]

6 Dec 2019

PONE-D-19-20413R1 

Availability, prices and affordability of essential medicines for treatment of diabetes and hypertension in private pharmacies in Zambia. 

Dear Dr. Forsberg:

I am pleased to inform you that your manuscript has been deemed suitable for publication in PLOS ONE. Congratulations! Your manuscript is now with our production department. 

With kind regards,

on behalf of

Dr. Khin Thet Wai 

Academic Editor

PLOS ONE